# On mechanisms for HF pump-enhanced optical emissions at 557.7 and 630.0 nm from atomic oxygen in the high-latitude F-region ionosphere

Thomas B. Leyser[1], Tima Sergienko[2], Urban Brändström[2], Björn Gustavsson[3], and Michael T. Rietveld[4]

[1]Swedish Institute of Space Physics, Uppsala, Sweden.
[2]Swedish Institute of Space Physics, Kiruna, Sweden.
[3]Department of Physics and Technology, The Arctic University of Norway, Tromsø, Norway.
[4]Scientist emeritus, EISCAT Scientific Association, Ramfjordmoen, Norway.

**Correspondence:** Thomas Leyser (thomas.leyser@irfu.se)

**Abstract.** The EISCAT (European Incoherent Scatter association) Heating facility was used to transmit powerful high frequency (HF) electromagnetic waves into the F-region ionosphere to enhance optical emissions at 557.7 and 630.0 nm from atomic oxygen. The emissions were imaged by several stations of ALIS (Auroral Large Imaging System) in northern Sweden and the EISCAT UHF incoherent scatter radar was used to obtain plasma parameter values. The ratio of the 557.7 to 630.0 nm column emission rates changed from $I_{5577}/I_{6300} \approx 0.2$ for the HF pump frequency $f_0 = 6.200$ MHz $\approx 4.6 f_{\rm e}$ to $I_{5577}/I_{6300} \approx 0.5$ when $f_0 = 5.423$ MHz $\lesssim 4 f_{\rm e}$, where $f_{\rm e}$ is the ionospheric electron gyro frequency. The observations are interpreted in terms of decreased electron heating efficiency and thereby weaker enhancement at 630.0 nm for $f_0 = 5.423$ MHz $\lesssim 4 f_{\rm e}$. The emissions at 557.7 nm are attributed to electron acceleration by upper hybrid waves of meter-scale wavelengths that can be excited with $f_0 = 5.423$ MHz $\lesssim 4 f_{\rm e}$.

## 1 Introduction

Powerful high frequency (HF) electromagnetic waves transmitted into the ionosphere from the ground may enhance optical emissions from atmospheric constituents, notably atomic oxygen and molecular nitrogen. Such emissions can be detected on the ground and are studied to get information on a variety of phenomena related to plasma energization by HF pumping. Following the first unambiguous observations of HF pump-enhanced optical emissions at high latitudes (Brändström et al., 1999; Kosch et al., 2002b), a number of interesting results have been obtained. Here we limit ourselves to HF pumping with the frequency $f_0$ below or near the ionospheric critical frequency and with left-handed circular polarization (often referred to as ordinary mode) which gives the strongest effects.

The spatial distribution of optical emissions enhanced above background levels has shown that the strongest coupling between an HF beam and the ionospheric F-region plasma at high latitudes and long time scales occurs for a beam in the direction of geomagnetic zenith. Even with a vertical beam, the most intense optical emissions occur towards magnetic zenith within the beam. This was observed in emissions at 630.0 nm from the O($^1$D) excited state as obtained with the EISCAT (European Incoherent Scatter association) Heating facility in Norway (Kosch et al., 2000), with the Sura facility in Russia (Grach et al., 2007;

Shindin et al., 2015), as well as with the HAARP (High Frequency Active Auroral Research Program) facility in Alaska, USA (Pedersen and Carlson, 2001), and also at 557.7 nm from $O(^1S)$ (Pedersen et al., 2003, 2008). In addition, optical emissions show evidence of complex HF beam re-organization during pumping and associated nonlinear effects in the pump–plasma interactions, as found in experiments both at EISCAT and HAARP (Kosch et al., 2004, 2007).

Imaging of enhanced optical emissions from several stations with ALIS (Auroral Large Imaging System) in northern Sweden and HF transmissions with EISCAT Heating provided for the first tomography-like reconstruction of the volume emission at 630.0 nm (Gustavsson et al., 2001). By comparing the obtained altitude and temporal variations of the optical emissions with modelled emissions of excitation of $O(^1D)$ by a maxwellian electron velocity distribution, the authors predicted the source distribution for the emissions to be maxwellian at low energies but with a depletion above 1.96 eV (which is the threshold for the excited state).

Observation of optical emissions simultaneously at several wavelengths, each requiring different minimum electron energy to be enhanced, provides the possibility to study electron energization in the excited plasma turbulence. Gustavsson et al. (2002) obtained nearly simultaneous images of enhancements at 557.7 and 630.0 nm in experiments with EISCAT Heating. The 557.7 to 630.0 nm intensity ratio of 0.3–0.4 was relatively high which implies that the optical enhancements were caused by a nonthermal tail in the electron velocity distribution. A similarly high ratio has been observed in experiments with HAARP (Pedersen et al., 2003).

Gustavsson and Eliasson (2008) observed emissions at 427.8, 557.7, 630.0 and 844.6 nm, together with incoherent scatter radar measurements of the ion temperature, electron temperature and concentration at EISCAT. By using a two-stream electron transport code they could estimate the electron flux between 1.9 and 100 eV that is needed to account for the observations. The electron energy distribution was found to be depleted approximately in the range 2–4 eV, probably due to excitation of vibrational states in $N_2$, and have a nonthermal tail at higher energies.

Experimental results on pump-enhanced optical emissions have been presented for varying $f_0$ near an harmonic $s$ ($s = 3$, 4) of the ionospheric electron gyro frequency $f_e$. Kosch et al. (2002a) observed simultaneously emissions at 630.0 nm and small-scale geomagnetic field-aligned density striations with the CUTLASS coherent scatter radar. They found a significant reduction in the optical enhancement and the radar backscatter when $f_0 \approx 3f_e$, indicating that the 630.0 nm emissions are linked to upper hybrid turbulence and associaated filamentary striations.

Further, Gustavsson et al. (2006) performed EISCAT experiments with $f_0$ stepping near $sf_e$ ($s = 3$, 4) and optical imaging at 630.0, 557.7 and 427.8 nm together with incoherent scatter radar measurements of the electron temperature. The pump-enhanced emissions as well as electron temperature were all minimum for $f_0 \approx 4f_e$. Whereas the enhancement of the emission intensity at 630.0 nm and electron temperature were roughly symmetric for $f_0$ around $4f_e$, the emissions at 427.8 nm were markedly stronger for $f_0$ a few tens of kilohertz above $sf_e$. This suggests that there are different electron energization processes underlying the emissions at 630.0 nm and 427.8 nm. The observations at 427.8 nm were the first direct evidence of pump-induced ionization of thermospheric $N_2$. The experimental results are consistent with theory that predict electron acceleration by upper hybrid oscillations localized in cylindrical density depletions and formation of a suprathermal tail in the electron

velocity distribution (Istomin and Leyser, 2003; Najmi et al., 2017). The acceleration efficiency is the largest for $f_0$ slightly above $s f_e$ for $s \geq 3$.

To further analyze these experimental results, Sergienko et al. (2012) performed Monte Carlo simulations of the transport of energized electrons into the ambient thermosphere. The observed 630.0 nm emissions could be accounted for by predominantly thermal electrons ($> 70\%$ of the emission intensity), with accelerated electrons playing a minor role for the emissions. However, in addition to the electron heating, to explain the observed intensity ratios for the different optical lines, electrons must be accelerated to 60 eV or more.

Shindin et al. (2015) observed emissions at 557.7 and 630.0 nm at the Sura facility for $f_0$ near $4 f_e$, an ERP of about 100 MW and the HF beam directed either vertically or tilted 12° south from the vertical in the magnetic meridional plane towards magnetic zenith (19° south). Simultaneous observations of the frequency spectrum of stimulated electromagnetic emissions were used to determine where $f_0$ was relative to $4 f_e$. The pump-enhanced 630.0 nm emissions did not exhibit a dependence on $f_0$ near $4 f_e$. And no minimum in the optical emission intensities was observed when $f_0 \approx 4 f_e$, which is contrary to previous measurements by others as mentioned above. The authors attribute this to natural variations in the ionospheric interaction altitudes during the experiments. Further, the authors found that for a vertical HF beam, 557.7 nm emissions occurred for $f_0$ about 5–15 kHz below $4 f_e$ and also at about 220–280 kHz above $4 f_e$. With the HF beam directed towards magnetic zenith, pump-enhanced emissions were observed for $f_0$ about 15–20 kHz above $4 f_e$.

In addition to these results for $f_0 \gtrsim 3 f_e$, a number of interesting experiments and theories have been presented for $f_0$ near $2 f_e$, but those are outside the scope of the present paper, as the relevant dispersion properties of electron Bernstein and upper hybrid modes are different for $s = 2$ and $s \geq 3$. Here we are concerned with $f_0$ near $4 f_e$ and higher.

We present experimental results from EISCAT Heating and optical imaging at 557.7 and 630.0 nm from atomic oxygen. Both the temporal evolution of the pump-enhanced emissions following pump-on and their intensity ratio were studied. With imaging from three ALIS sites, tomography-like reconstruction of the volume emission rates provided altitude profiles that under some assumptions imply a limit on the wavelength range of upper hybrid waves believed to be instrumental in the mechanism that causes the optical enhancements. Specifically, we discriminate conditions for the two different optical lines for $f_0$ near and well above $4 f_e$.

## 2 Experiment setup

The EISCAT Heating facility (Rietveld et al., 2016) was used to transmit HF waves in the cycle 150 s on/85 s off with the beam pointing in geomagnetic zenith ($\sim 78°$ elevation south) and with left-handed circular polarization (ordinary mode; "left-handed" is with respect to the direction of the geomagnetic field), on 16 February 2015. From 16:00:00 to 16:49:30 UT, $f_0 = 6.200$ MHz and the effective radiative power (ERP) was approximately 138 MW. To keep $f_0$ below the decreasing ionospheric ordinary-mode critical frequency $foF2$ after sunset, $f_0 = 5.423$ MHz from 16:50:55 to 17:09:05 UT and the ERP was 116 MW. The difference in the ERP is mainly due to different antenna gains at the different $f_0$. Local time was UT+1 hour.

Optical emissions observed with the ALIS stations in Abisko, Kiruna and Tjautjas were analyzed. The cameras had a field of view of $60°$ and gave an image size of $512 \times 512$ pixels with a temporal resolution of 7 s. In the data analysis, no additional integration was done apart from the exposure time. Emissions were imaged at 427.8 nm from $N_2^+$(1NG) with the threshold 18.6 eV (Holma et al., 2006), 557.7 nm from $O(^1S)$ with the threshold 4.17 eV (Haslett and Megill, 1974), 630.0 nm from $O(^1D)$ with the threshold 1.96 eV (Haslett and Megill, 1974), and 844.6 nm from $O(3p^3P)$ with the threshold 10.99 eV (Gustavsson et al., 2005). However, in the present treatment we focus on the 557.7 and 630.0 nm lines which are the strongest and for which we have the most data, thereby minimizing uncertainties. Kvammen et al. (2019) analyzed all four emission lines for this same experiment and presented tomography-like reconstructions of the three-dimensional distribution of the optical volume emission rates. We too employed tomography-like inversion to get altitude profiles of the volume emission rates for the two spectral lines, by a method similar to that described by Gustavsson et al. (2001).

To get the pump-induced enhancements of the optical lines, the background nightglow needs to be subtracted from the total column emissions measured. The background was estimated by cutting out the well-defined pump-enhanced blob in the images, and then estimating the background nightglow in that area by linear interpolation pixel by pixel from the nightglow outside the pump-enhanced region. This background was then subtracted from the measured intensities.

The EISCAT UHF incoherent scatter radar, transmitting 1.3–1.4 MW at about 930 MHz, was operated with the parabolic dish antenna scanning in a meridional pattern to obtain background plasma parameter values at different angles to the geomagnetic field, employing the beata program with a temporal resolution of 5 s and altitude resolution of approximately 3 km. Figure 1 shows the geometry of the experiment with the relative positions of the EISCAT facilities at the Ramfjordmoen site outside Tromsø, Norway, and the ALIS sites in the Kiruna region, Sweden.

Stimulated electromagnetic emission were received on the ground in Kroken (Tromsø), but were weak and only rarely exhibited spectral structure. Ionograms from the EISCAT Dynasonde show elevated D-region absorption.

## 3 Experimental results

Figure 2 shows keograms of the altitude profiles of pump-enhanced optical emissions at 630.0 nm above background as observed from the three ALIS sites during the experiment. The column emission rate weakened as $f_0$ was changed from 6.200 MHz to 5.423 MHz at about 16:50 UT. For the second HF pulse at $f_0 = 5.423$ MHz observed at the Abisko station, the relatively short duration shown in the enhancement is due to lacking synchronization between the pump cycle and optical filter wheel sequence. In the keograms, the slow growth of the emission intensity and extension towards higher altitudes during the pump pulses can be seen. Also, the emissions occurred at lower altitudes for the lower $f_0$. Further, an altitude oscillation of the optical enhancements occurred with an amplitude of $\sim 30$ km and minima at approximately 16:28 and 16:58 UT, implying a period of $\sim 30$ min. This is presumably an atmospheric gravity wave and associated traveling ionospheric disturbance (e.g., Hunsucker, 1982). The altitude decrease as $f_0$ was lowered is likely a combination of this ionospheric oscillation and that an HF wave with a lower $f_0$ will reflect at a lower altitude because the plasma density increases with height in the bottom-side ionosphere.

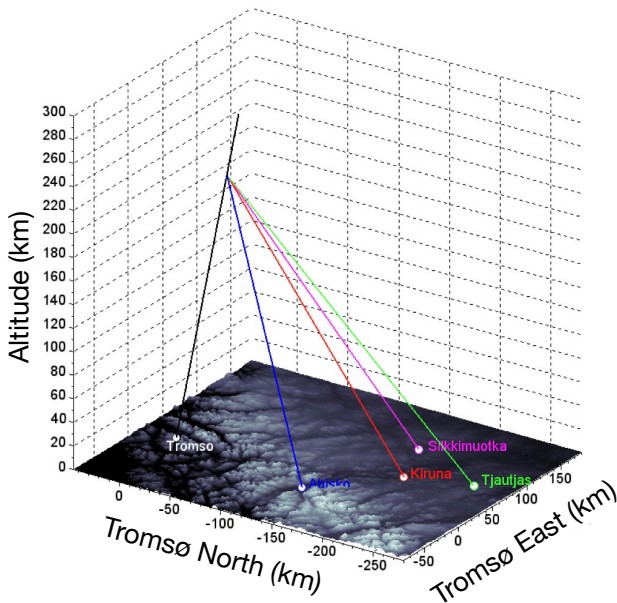

**Figure 1.** Illustration of the relative positions of the EISCAT facilities at the Ramfjordmoen site outside Tromsø, Norway, and the ALIS stations in the Kiruna region, Sweden. The black line shows the direction of magnetic zenith. Data from the ALIS site at Silkkimuotka was not used in the analysis.

Figure 3 displays similarly an overview of the enhanced emissions at 557.7 nm for the same time period. The emission intensity did not weaken as $f_0$ was decreased at about 16:50 UT. The images from the Kiruna site show even an increase in
the emissions as $f_0$ was decreased, despite a lower ERP. But again the altitude region of the emissions decreased as $f_0$ was lowered. It is interesting that Kvammen et al. (2019) observed a clear intensity increase in the 844.6 nm line which has a threshold of 10.99 eV, which similarly to the threshold of 4.17 eV of the 557.7 nm line is several times above the threshold of 1.96 eV for 630.0 nm.

Figure 4 shows the altitude profiles of the electron density, electron temperature and ion temperature obtained with the
EISCAT UHF radar in geomagnetic zenith. Clear pump-induced enhancements in the electron temperature can be seen which are slowly conducted upward in the ionosphere with time, along the geomagnetic field where the thermal conductivity is the highest. A similar slow upward motion can be seen in the 630.0 nm data (Fig. 2). The electron temperature reaches above 3000 K. The electron density and ion temperature did not exhibit notable pump-induced modulations. However, the electron density slowly decreased with time as the experiments occurred shortly after sunset.

Figure 5 shows the altitude profiles of the volume emission rates at 630.0 nm (left) and 557.7 nm (right) for the last pump cycle at $f_0 = 6.200$ MHz, obtained by tomography-like reconstruction of the emission regions. The maximum volume emission rates occurred at approximately the same height for the two spectral lines, near $h_{max} \approx 239$ km. The altitude profile for the 557.7 nm emission is wider than that for 630.0 nm, which is consistent with that the excitation threshold for the former

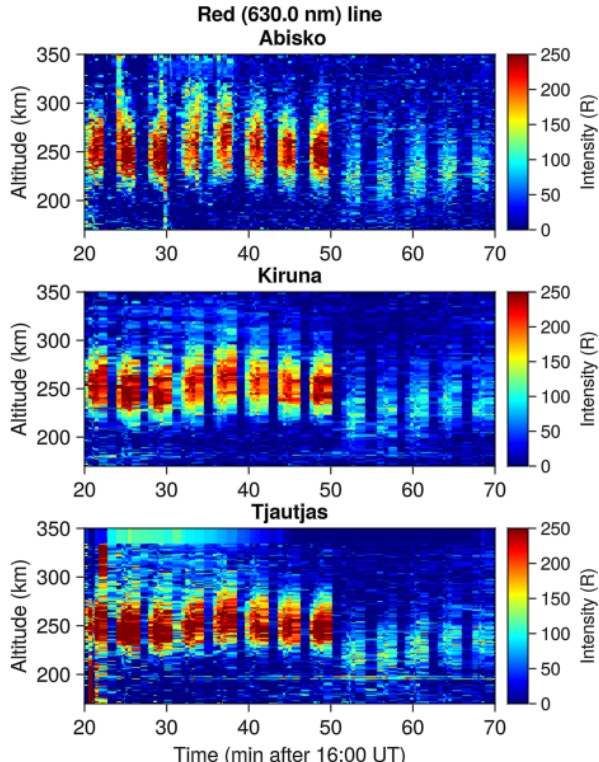

**Figure 2.** Keograms of pump-enhanced column emissions above background at 630.0 nm from 16:20:00 to 17:09:05 UT on 16 February 2015. From 16:00:00 to 16:49:30 UT, $f_0 = 6.200$ MHz, and from 16:50:55 to 17:09:05 UT, $f_0 = 5.423$ MHz.

(4.17 eV) is higher than for the latter (1.96 eV). Electrons with higher energies will have longer mean free paths, so that the volume within which they collide with neutrals and excite optical emissions will be larger. The times for the plotted profiles are determined by the sequence in the filter wheels for the ALIS cameras. Figure 6 shows similarly the height profiles of the volume emission rate for the third pump pulse at $f_0 = 5.423$ MHz. In this case, the maximum of the emissions at 557.7 nm (right) occurred about 7–10 km below that at 630.0 nm (left). Also, by comparing Figs. 5 and 6, we see that the emission rates for both lines grow slower with $f_0 = 5.423$ MHz.

Figure 7 summarizes the temporal evolution of the column emission rates at 630.0 nm (left) and 557.7 nm (right) during the growth phase following pump-on for $f_0 = 6.200$ MHz (red) and $f_0 = 5.423$ MHz (blue), for the same data as in Figs. 5 and 6. The 630.0 nm emissions are seen to be weaker for $f_0 = 5.423$ MHz than for $f_0 = 6.200$ MHz. However, the emission intensity at 557.7 nm is approximately the same for the two $f_0$ towards the end of the pump pulses. The intensity ratio towards the end of the plotted time interval is $I_{5577}/I_{6300} \approx 0.2$ for $f_0 = 6.200$ MHz and $I_{5577}/I_{6300} \approx 0.5$ for $f_0 = 5.423$ MHz, which agrees with Kvammen et al. (2019) for the same experiment. We do not have data for the remaining part of the pump pulse of 150 s because of the used sequence for the filter wheels.

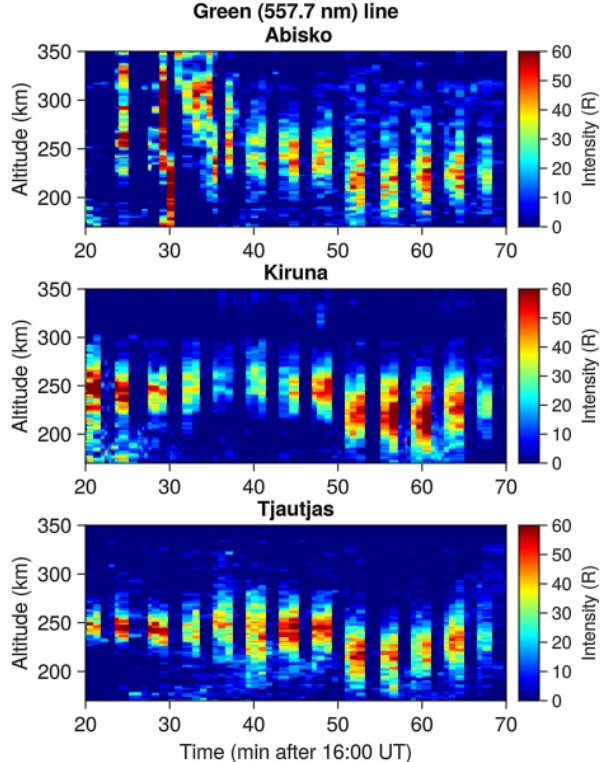

**Figure 3.** The same as Fig. 2, but for the enhancements at 557.7 nm.

## 4 Emission growth times

To obtain a measure of the growth time of the optical intensities after pump-on, we fit a theoretical growth time to the experimental data for the emission from a given excited state. The temporal evolution of an emission can be described by the continuity equation for the number density $n_\alpha$ of atoms in the excited state $\alpha$ (O($^1$D), O($^1$S)). $n_\alpha$ is related to the observed column emission rate $I_\lambda$ through $n_\alpha(t) = I_\lambda(t)/A_\alpha/D_\lambda(t)$, where $A_\alpha$ is the Einstein coefficient(s) for the emission and $D_\lambda$ is the observed spatial scale of the imaged optical region at the wavelength $\lambda$. The number density $n_\alpha$ is described by the equation

$$\frac{dn_\alpha}{dt} + \nabla \cdot (n_\alpha \mathbf{v}) = Q_\alpha(\mathbf{r}, t) - L_\alpha(t) n_\alpha \tag{1}$$

where $Q_\alpha$ is the excitation rate which includes the electron impact excitation (thermal and accelerated) and chemical reactions. The loss rate $L_\alpha$ is given by

$$L_\alpha(t) = \sum_i (q_i n_i) - \sum_j A_j \tag{2}$$

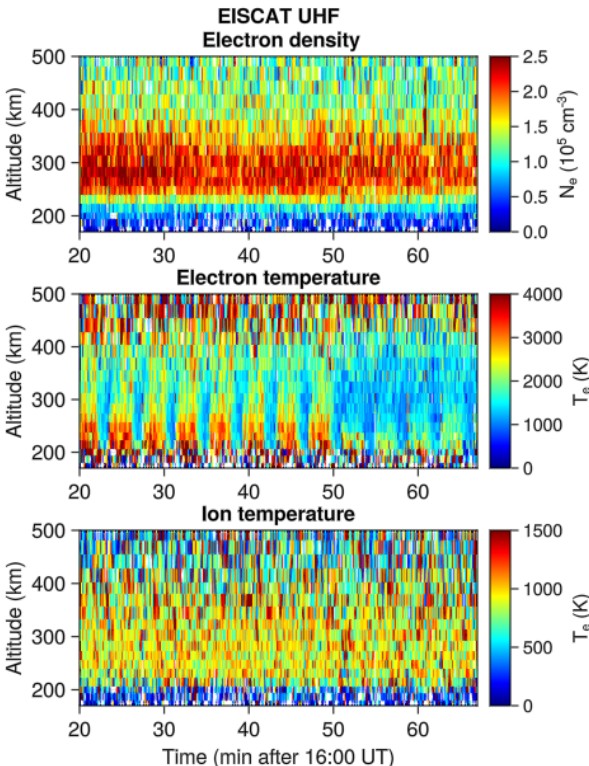

**Figure 4.** Overview of the electron density (top), electron temperature (center) and ion temperature (bottom) as obtained with the EISCAT UHF radar directed in magnetic zenith, from 16:20:00 to 17:09:05 UT on 16 February 2015. From 16:00:00 to 16:49:30 UT, $f_0 = 6.200$ MHz and from 16:50:55 to 17:09:05 UT, $f_0 = 5.423$ MHz.

where $q_i$ is the rate coefficient for collisional de-excitation (quenching) by collisions with other species with the number density $n_i$. The last term is the sum of the Einstein coefficients for all emissions originating from the excited state in question. For the 165   630.0 nm emission, $L_{\mathrm{O}(^1\mathrm{D})}$ includes the sum of three Einstein coefficients and $q_i$ is for reactions with the main neutrals in the thermosphere at the heights of the F-region ionosphere ($N_2$, $O_2$ and O) as well as with thermal electrons. For the 557.7 nm line, $L_{\mathrm{O}(^1\mathrm{S})}$ is the sum of the Einstein coefficients only since the $\mathrm{O}(^1\mathrm{S})$ state is not quenched by collisions at the F-region altitudes of the experiments. Furthermore, for the experiment conditions, the term $\nabla \cdot (n_\alpha \mathbf{v})$ in Eq. (1) can be neglected because of the low velocity $\mathbf{v}$ of the neutrals. For the purpose of the present analysis, the spatial dependencies of $Q_\alpha$ and $L_\alpha$ are neglected as 170   we are interested only in a relatively small spatial region around the intensity maximum, not in the entire emitting volume.

For our case of HF pump-enhanced emissions, the loss rate is constant with $L_\alpha = 1/\tau_\alpha$, where $\tau_\alpha$ is the effective lifetime of the excited state $\alpha$. For the $\mathrm{O}(^1\mathrm{S})$ state, $\tau_{\mathrm{O}(^1\mathrm{S})}$ equals the radiative lifetime so that $\tau_{\mathrm{O}(^1\mathrm{S})} \approx 1.3$ s, because this state is not quenched by collisions. For the $\mathrm{O}(^1\mathrm{D})$ state, $\tau_{\mathrm{O}(^1\mathrm{D})}$ is affected by quenching and we take $\tau_{\mathrm{O}(^1\mathrm{D})}$ for the relevant altitudes from Gustavsson et al. (2001).

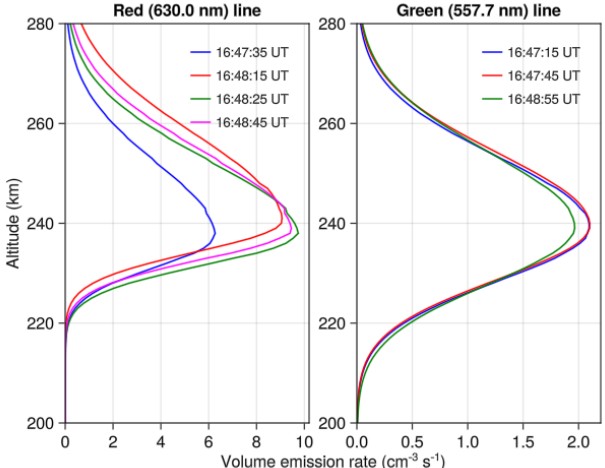

**Figure 5.** Temporal evolution of the height profile of the pump-enhanced volume emission rates in magnetic zenith at 630.0 nm (left) and 557.7 nm (right) for $f_0 = 6.200$ MHz. The accuracy of the height determination is approximately $\pm 1$ km.

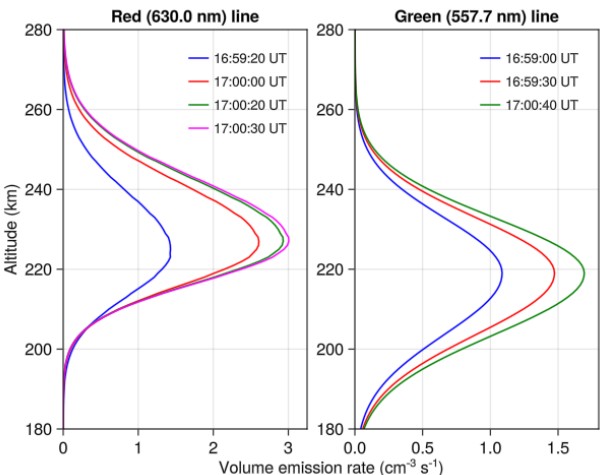

**Figure 6.** The same as Fig. 5, but for $f_0 = 5.423$ MHz.

For the purpose of our analysis, we take the source term $Q_\alpha$ to be constant. The solution to Eq. (1) is

$$n_\alpha(t) = \tau Q_\alpha \left[ 1 - \exp\left( -\frac{t}{\tau} \right) \right] \qquad (3)$$

With $Q_\alpha$ constant, the growth time is $\tau = \tau_\alpha$. However, for the general case when $Q_\alpha$ is time dependent, $\tau$ is determined by both the lifetime $\tau_\alpha$ and $Q_\alpha(t)$. Therefore, if fitting the experimental data by Eq. (3) for which $Q_\alpha$ is constant, gives an observed growth time $\tau_{\mathrm{obs}}$ very different from $\tau_\alpha$, this implies that the source $Q_\alpha$ is not constant.

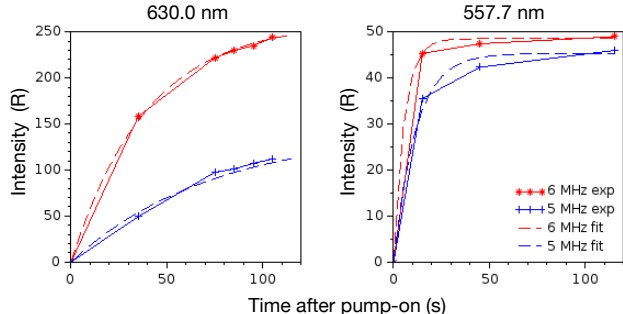

**Figure 7.** Temporal evolution of the pump-enhanced column emission rates at 630.0 nm (left) and 557.7 nm (right) after pump-on at time $t = 0$ s, for the same pump pulses as in Figs. 5 and 6. The solid lines connect the data points shown by markers (red $*$ for $f_0 = 6.200$ MHz and blue $+$ for $f_0 = 5.423$ MHz). The dashed lines are the fits with Eq. (3) which give $\tau_{\text{obs}}$.

**Table 1.** Emission growth times $\tau_{\text{obs}}$ after pump-on obtained by fitting Eq. (3) with the observed column emission rates at 630.0 nm ($I_{6300}$) and 557.7 nm ($I_{5577}$). $\tau_\alpha$ is the theoretical effective lifetime of the excited state $\alpha$.

| $I_\lambda$ | $f_0$ (MHz) | $\tau_\alpha$ (s) | $\tau_{\text{obs}}$ (s) |
|---|---|---|---|
| $I_{6300}$ | 6.200 | 40 | 39 |
| $I_{5577}$ | 6.200 | 1.3 | 5.4 |
| $I_{6300}$ | 5.423 | 25 | 74 |
| $I_{5577}$ | 5.423 | 1.3 | 11 |

To obtain $\tau_{\text{obs}}$ from the measurements, we recall that $n_\alpha(t) = I_\lambda(t)/A_\alpha/D_\lambda(t)$, so that the observed intensity $I_\lambda(t) \propto n_\alpha(t)$ for a fixed $D_\lambda$. Thus, $I_\lambda(t)$ is fitted with a quantity proportional to Eq. (3) for the images at the different $t = t_k$ of the data. $I_\lambda(t_k)$ is averaged over a small square ($15 \times 15$ pixels) around the maximum intensity in the images. The fitting was done by the least squares Levenberg–Marquardt algorithm.

Table 1 summarizes $\tau_{\text{obs}}$ for the column emission rates at 557.7 and 630.0 nm. The theoretical lifetime $\tau_{\text{O}(^1\text{D})} \approx 40$ s in the third column from the left, is for the altitude $h_{\max} \approx 239$ km (Gustavsson et al., 2001) where the 630.0 nm volume emission rate is maximum for $f_0 = 6.200$ MHz (Fig. 5). Similarly, $\tau_{\text{O}(^1\text{D})} \approx 25$ s is for $h_{\max} \approx 225$ km (Gustavsson et al., 2001) with $f_0 = 5.423$ MHz (Fig. 6).

With $f_0 = 6.200$ MHz, $\tau_{\text{obs}}$ for the 630.0 nm line is almost equal to $\tau_{\text{O}(^1\text{D})}$. For the 557.7 nm line, $\tau_{\text{obs}}$ is approximately four times larger than $\tau_{\text{O}(^1\text{S})}$. These results suggest that for $f_0 = 6.200$ MHz, the excitation sources $Q_\alpha$ for both emissions reach their maximum faster than $\tau_{\text{O}(^1\text{D})} \approx 40$ s but slower than $\tau_{\text{O}(^1\text{S})} \approx 1.3$ s, and probably the source growth rates are different for the two emissions.

On the other hand, with $f_0 = 5.423$ MHz, $\tau_{\text{obs}}$ for both emissions is several times larger than $\tau_\alpha$. In addition, $\tau_{\text{obs}}$ for both emissions is larger than with $f_0 = 6.200$ MHz. This suggests that for $f_0 = 5.423$ MHz, the source $Q_\alpha$ increases during

pumping, and slower than for $f_0 = 6.200$ MHz. A relatively slow growth of emissions at 557.7 nm has also been observed in experiments at the HAARP facility with $f_0 = 7.8$ MHz and an ERP of approximately 160 MW (Pedersen et al., 2003). The 557.7 to 630.0 nm intensity ratio in that case was approximately 1:3.

## 5   Discussion

HF pump-excitation of the O($^1$D) state, the source of the 630.0 nm line, has been attributed to mainly electron heating from a maxwellian electron distribution (Mantas, 1994; Mantas and Carlson, 1996), however, taking into account collisional de-excitation by collisions with molecular oxygen and nitrogen in the atmosphere (Gustavsson et al., 2001). The association of the 630.0 nm emissions with electron heating is consistent with our results for the optical (Fig. 2) and electron temperature enhancements (Fig. 4). Both decreased when $f_0$ was changed from 6.200 to 5.423 MHz at about 16:51 UT.

One feature of the results of Klimenko et al. (2017) attributed observations of pump-enhanced emissions at 630.0 nm to electron acceleration rather than heating, for experiments at the Sura facility. This conclusion was arrived at by modeling the contributions to the 630.0 nm emissions from dissociative recombination of $O_2^+$ ions and electrons, thermal electron heating and acceleration. The best fit of modeled and experimental results were obtained with moderate electron temperatures of 1100–1900 K. Further, Klimenko et al. (2017) found that the 630.0 nm emissions would be mainly due to electron heating for electron temperatures above about 2500 K. This is consistent with our results and conclusion of the importance of heating as we measured electron temperatures above 3000 K (Fig. 4). And this was obtained with an ERP of $\sim 138$ MW and $\sim 116$ MW, compared to the $\sim 100$ MW in the Sura experiments.

One feature of the results of Klimenko et al. (2017) was that no enhancements were observed when $foF2 \lesssim f_0 + 0.5$ MHz. In our case, $foF2 \lesssim f_0 + 0.5$ MHz throughout the experiment, as obtained from the EISCAT Dynasonde. This difference in conditions for pump-enhancements may possibly be related to the different latitudes of the experiments, with EISCAT at high latitude within the auroral zone and Sura at mid latitude with a smaller dip angle of the geomagnetic field. The dip angle can influence the importance of HF pump excitation parallel to the geomagnetic field (Langmuir turbulence) relative to perpendicular to the magnetic field (electron Bernstein and upper hybrid phenomena).

HF pump-enhanced optical emissions have been connected to upper hybrid waves because the emission intensities are sensitive to $f_0$ near $sf_e$ (Kosch et al., 2002a; Gustavsson et al., 2006). The response of ionospheric F-region plasma to HF pumping is asymmetric around $sf_e$ (Leyser et al., 1989; Stubbe et al., 1994; Honary et al., 1995; Gustavsson et al., 2006). This is related to asymmetries in the dispersion characteristics of upper hybrid and electron Bernstein modes for frequencies near $sf_e$ and $s \geq 3$. Theories of the localization of upper hybrid oscillations in density depletions of filamentary density striations along the geomagnetic field predict an asymmetry in the trapping mechanism with deeper depletions for $f_0 \gtrsim sf_e$ than for $f_0 \lesssim sf_e$ (Mjølhus, 1993). Therefore stronger pump-induced effects are expected for $f_0 \gtrsim sf_e$ than for $f_0 \lesssim sf_e$ which gives a corresponding asymmetry also in the anomalous absorption of HF waves by scattering on the striations (Honary et al., 1995; Gurevich et al., 1996) and electron acceleration by the localized upper hybrid oscillations (Istomin and Leyser, 2003). These phenomena are expected to be correlated with electron temperature enhancements which are the source of optical emissions at

630.0 nm and which therefore too are expected to be stronger for $f_0 \gtrsim s f_e$ than for $f_0 \lesssim s f_e$. With $f_0 \approx s f_e$, upper hybrid and electron Bernstein phenomena are suppressed because of the linear dispersion characteristics.

In order to find out the relation between $f_0 = 5.423$ MHz and $4 f_e$ in our experiment, we use the IGRF model (International Geomagnetic Reference Field, 13th generation) for the altitude variation of the geomagnetic field in the pump–ionosphere interaction region. We find that $4 f_e \approx f_0 = 5.423$ MHz occurred at the height of $h \approx 235$ km for the date of our experiment. This altitude is above that for the maximum volume emission rates at both 630.0 and 557.7 nm. Figure 6 shows this height to be $h_{\max} \approx 227$ km for 630.0 nm and $h_{\max} \approx 219$ km for 557.7 nm. Since $f_e$ increases with decreasing altitude, we conclude that $f_0 < 4 f_e$ in the regions with maximum volume emission rates. The IGRF model gives for the 630.0 nm emissions, $f_e \approx 1.360$ MHz at $h_{\max} \approx 227$ km, so that $\Delta f \equiv f_0 - 4 f_e \approx -17$ kHz. For the emissions at 557.7 nm, $f_e \approx 1.364$ MHz at $h_{\max} \approx 219$ km, giving $\Delta f \approx -33$ kHz.

Stimulated electromagnetic emissions can also be used to estimate the vicinity of $f_0$ to $s f_e$ (Leyser, 2001). In our experiment the emissions were generally too weak for spectral structure to be identified. However, for the fourth pump pulse at $f_0 = 5.423$ MHz, a weak broad upshifted maximum (BUM) can be observed. Figure 4 shows the electron temperature to be slightly more enhanced in the fourth (and last) pump pulse than in the preceding pulses at $f_0 = 5.423$ MHz, which indicates a stronger plasma excitation in the last pulse. Figure 8 displays a 130 kHz wide spectrum of the stimulated electromagnetic emissions in the top panel with the ionospherically reflected pump wave at $f_0$ and, for comparison, the noise level with interfering HF transmissions in the same frequency range during pump-off in the bottom panel. The BUM has its maximum at $f_{BUM} \approx f_0 + 15$–20 kHz, which is near the so called cutoff frequency of the BUM (Leyser, 2001) that is the minimum frequency upshift of the BUM from $f_0$ below which the BUM is not excited. For $s = 4$, the BUM has been observed in the range $4 f_e - 20$ kHz $\lesssim f_0 \lesssim 4 f_e + 120$ kHz (Frolov et al., 1998), thus mainly for $f_0 \gtrsim 4 f_e$. However, the proximity of $f_{BUM}$ to the cutoff frequency indicates that $f_0 \approx 4 f_e$ in the excitation region. The fact that no downshifted maximum (DM) is visible at about $f_0 - 10$ kHz is another indication of that $f_0 \approx 4 f_e$. It may be that the optical emissions and the BUM emissions were excited at different altitudes with slightly different $f_e$. Presumably the weak BUM could have been excited at higher altitudes where $f_0 \approx 4 f_e$ and the optical emissions at lower altitudes where $f_0 \lesssim 4 f_e$. Also, we note that the height profiles in Figs. 5 and 6 are for the third pump pulse for which the quality of the optical data is higher than for the fourth pulse. The pump–plasma interaction altitudes may have increased slightly between the two pulses, due to the decreasing solar irradiation of the ionosphere after sunset.

Gustavsson and Eliasson (2008) discussed pump-enhanced optical emissions at 427.8, 557.7, 630.0 and 844.6 nm for $f_0 = 5.423$ MHz and cycling the pump wave 4 min on/2 min off. By fitting observed optical emissions and modelled enhanced electron fluxes they obtained estimates of the volume emission rates for the different optical lines. They found that the maximum volume emission rate for all spectral lines occurred at approximately the same altitude (their figure 7), which agrees with our observations for $f_0 = 6.200$ MHz (Fig. 5). Also, the ratio of the column emission rates for the emissions at 557.7 and 630.0 nm varied in the range $I_{5577}/I_{6300} \approx 0.1$–0.4 during the experiment (from their figure 2), which agrees with our ratio for the same emission lines $I_{5577}/I_{6300} \approx 0.2$, but again at $f_0 = 6.200$ MHz. In our case with $f_0 = 5.423$ MHz, the maximum of the emissions at 557.7 nm occurred about 7–10 km below that at 630.0 nm (Fig. 6). Further, the ratio $I_{5577}/I_{6300} \approx 0.5$ is slightly higher than that for the same emission lines presented by Gustavsson and Eliasson (2008) for $f_0 = 5.423$ MHz.

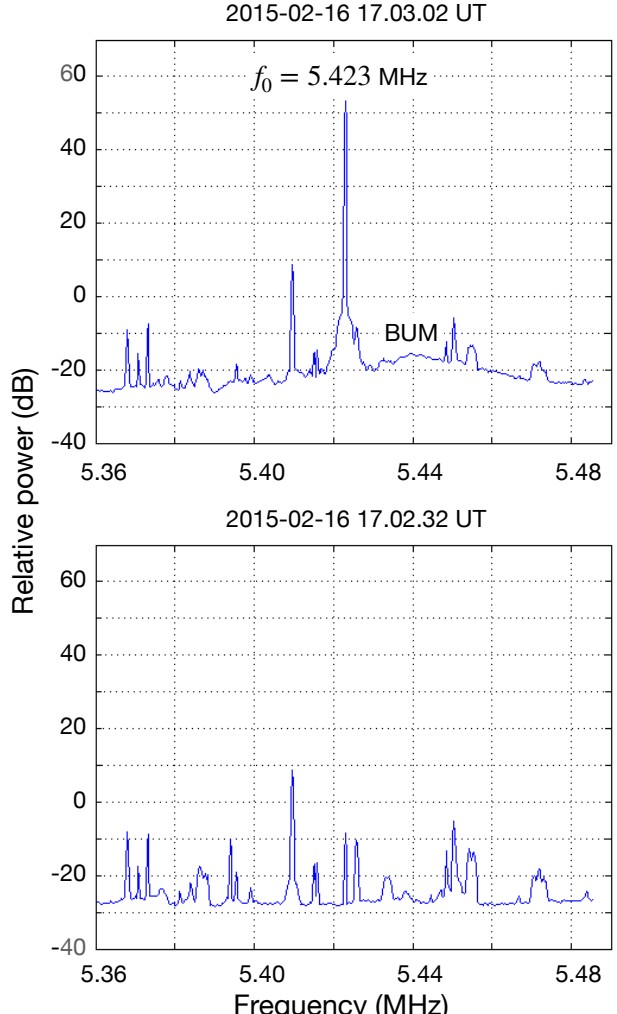

**Figure 8.** Stimulated electromagnetic emissions observed with $f_0 = 5.423$ MHz at 17:03:02 UT (top panel) and noise level with pump-off at 17:02:32 UT. A weak BUM with its peak intensity upshifted from $f_0$ by 15–20 kHz can be identified.

Gustavsson and Eliasson (2008) found the maximum volume emission rate for 630.0 and 557.7 nm at $h_{max} \approx 220$ km and higher, depending on time in the experiment (their figure 7). According to the IGRF model for the date of their experiments, $4f_e \approx 5.414$ MHz at 220 km and lower at increasing altitudes. Thus, in their case $f_0 \gtrsim 4f_e$ in the height region with the main pump-enhanced optical emissions. In view of the similarities between the results of Gustavsson and Eliasson (2008) and our case with $f_0 = 6.200$ MHz, together with that in our case $f_0 = 5.423$ MHz $\lesssim 4f_e$ at the altitudes of maximum volume emission rates as discussed above, we conjecture that $f_0 = 5.423$ MHz $\lesssim 4f_e$ in the energization regions where the optical emissions were enhanced in our experiment.

Further, the dispersion characteristics of upper hybrid waves (Leyser et al., 1989; Istomin and Leyser, 1995; Mishin et al., 2005) are similar for the case of Gustavsson and Eliasson (2008) with $f_0 = 5.423$ MHz $\gtrsim 4f_e$ and our case for $f_0 = 6.200$ MHz $\approx 4.6f_e$. In both cases, upper hybrid waves can have a wide range of wave numbers, which facilitates large electron temperature enhancements and emissions at 630.0 nm. On the other hand, for $f_0 \lesssim 4f_e$, as in our case for $f_0 = 5.423$ MHz, upper hybrid waves with positive group dispersion are limited to relatively small wave numbers only which is expected to give less electron heating and lower emission levels at 630.0 nm. Therefore, our result of $I_{5577}/I_{6300} \approx 0.5$ appears consistent with $I_{5577}/I_{6300} \approx 0.1$–0.4 as observed by Gustavsson and Eliasson (2008), both with $f_0 = 5.423$ MHz but the former with $f_0 \lesssim 4f_e$ and the latter with $f_0 \gtrsim 4f_e$.

The fact that the column emission rate at 630.0 nm in our case decreased when $f_0$ was lowered from $f_0 = 6.200$ to $f_0 = 5.423$ MHz is consistent with the theoretically predicted asymmetry in pump-induced effects for $f_0$ around $4f_e$, with weaker enhancements for $f_0 \lesssim 4f_e$ and the largest excitations for $f_0 \gtrsim 4f_e$. For $f_0 = 6.200$ MHz $\approx 4.6f_e$, the excitation level should be somewhere in between these two cases, that is, stronger than for $f_0 = 5.423$ MHz $\lesssim 4f_e$, as observed. However, we note that no such asymmetry in the 630.0 nm emission has been observed in previous experiments (Gustavsson et al., 2006; Shindin et al., 2015), although the emission intensity was minimum for $f_0 \approx sf_e$.

Emission at 557.7 nm has been interpreted in terms of electron acceleration giving a non-thermal tail in the electron distribution, rather than electron heating (Gustavsson et al., 2002, 2005). Istomin and Leyser (2003) presented a model of electron acceleration by upper hybrid oscillations localized in the density depletion of a striation pumped by a left-handed circularly polarized electromagnetic wave, that gives a power-law tail in the electron velocity distribution. From their equation (40) we obtain an estimate of the maximum upper hybrid wave number perpendicular to the geomagnetic field for $\Delta f \approx -33$ kHz (557.7 nm) as $k_{\perp \max}\rho_e \approx 0.1$, where $\rho_e$ is the thermal electron gyro radius. With an electron temperature of 2500 K (Fig. 4), $\rho_e \approx 0.023$ m, which gives $k_{\perp \max} \approx 4$ m$^{-1}$ or upper hybrid wavelengths longer than about 1 m. For $\Delta f \approx -17$ kHz (630.0 nm), $k_{\perp \max}$ is even smaller which implies an even smaller range of possible upper hybrid wavelengths. For wave numbers $k_\perp \gtrsim k_{\perp \max}$, upper hybrid waves have negative group dispersion for which the waves cannot be localized in density depletions and therefore can only be relatively weakly excited.

As the 557.7 nm column emission rate in our case was approximately the same or even increased as $f_0$ was decreased, we conclude that this emission is excited through electron acceleration by relatively long upper hybrid wavelengths, about 1 m or longer that can be excited for $f_0 = 5.423$ MHz $\lesssim 4f_e$ as well as for $f_0 = 6.200$ MHz. This assumes that the pump-induced electron energization occurred approximately at the altitudes where the optical volume emission rates are the largest. The slow growth of the 557.7 nm line relative to the lifetime of the O($^1$S) state (Table 1) is consistent with the fact that upper hybrid waves need to be localized in the slowly growing density depletions of striations for efficient electron acceleration. The fact that both the 630.0 and 557.7 nm emissions exhibited a much longer growth time than the lifetime of the corresponding excited states is consistent with the relatively weak level of upper hybrid phenomena that is expected for $f_0 \lesssim 4f_e$ as only relatively long wavelengths can be excited.

The emission intensity at 557.7 nm for $f_0 = 5.423$ MHz was relatively high despite the slightly lower ERP and the expected limited wavelength range of upper hybrid oscillations. This can be understood by the assumption that the anomalous absorption

of the pump wave (and associated electron temperature enhancements) decreased as $f_0$ was lowered from 6.200 MHz to 5.423 MHz, thereby giving stronger pumping of the long-wavelength upper hybrid oscillations for electron acceleration. Conversely, the larger enhancements of both the 630.0 nm intensity and the electron temperature for $f_0 = 6.200$ MHz are attributed to mainly the wider range of upper hybrid wavelengths that are available compared to that for $f_0 = 5.423$ MHz $\lesssim 4f_{\mathrm{e}}$, in addition to the slightly higher ERP.

Kvammen et al. (2019) concluded for the same experiment that $f_0 = 5.423$ MHz $\gtrsim 4f_{\mathrm{e}}$ by comparing their observed increase in the enhancement at 844.6 nm (10.99 eV threshold) for $f_0 = 5.423$ MHz to Gustavsson et al. (2006) who observed larger enhancements at 427.8 nm (18.75 eV threshold) for $f_0 \gtrsim 4f_{\mathrm{e}}$ than for $f_0 \lesssim 4f_{\mathrm{e}}$. We note that the ERP in the experiments by Gustavsson et al. (2006) was four to five times higher than in the present experiment, implying that a comparison between the experiments may not be obvious. In view of that $f_0 = 5.423$ MHz $\lesssim 4f_{\mathrm{e}}$ at the altitudes of maximum volume emission rates we suggest the possibility that $f_0 \lesssim 4f_{\mathrm{e}}$ in the excitation regions of the optical emissions. With $f_0 \lesssim 4f_{\mathrm{e}}$ at lower altitudes and $f_0 \gtrsim 4f_{\mathrm{e}}$ at higher altitudes, there would be a region with $f_0 \approx 4f_{\mathrm{e}}$ in between with minimum pump–plasma interaction involving upper hybrid and electron Bernstein modes, therefore minimum electron heating and enhancements at 630.0 nm. With $f_0$ sufficiently near $4f_{\mathrm{e}}$, regardless of whether $f_0 \lesssim 4f_{\mathrm{e}}$ or $f_0 \gtrsim 4f_{\mathrm{e}}$, upper hybrid and electron Bernstein oscillations would have only relatively long wavelengths, which are suggested to be involved in the electron acceleration that enhances emissions at 557.7 nm (and 844.6 nm as studied by Kvammen et al. (2019)).

Najmi et al. (2017) presented results from Vlasov and test-particle simulations that indicate the physics of electron energization by wave modes perpendicular to the ambient magnetic field and trapped in the density depletion of a striation in the upper hybrid resonance region. When $f_0$ is between $3f_{\mathrm{e}}$ and $4f_{\mathrm{e}}$, bulk electron heating occurs by electron Bernstein waves which leads to an essentially thermal electron velocity distribution. For $f_0 \gtrsim 4f_{\mathrm{e}}$, resonant interaction with the upper hybrid mode was found to dominate which leads to electron acceleration and the formation of a suprathermal tail of energetic electrons. However, with $f_0 = 4.01 f_{\mathrm{e}}$ in their simulations, the frequency shift of $f_0$ above $4f_{\mathrm{e}}$ is less than the lower hybrid frequency they used. This implies that decay products of parametric interaction between upper hybrid oscillations at $f_0$ and lower hybrid oscillations have frequencies below $4f_{\mathrm{e}}$, so that the interaction involves upper hybrid waves both slightly above and below $4f_{\mathrm{e}}$. Their results can therefore not be used for a comparison of electron energization efficiency of $f_0 \lesssim 4f_{\mathrm{e}}$ and $f_0 \gtrsim 4f_{\mathrm{e}}$ as would be relevant for the present study. But the importance of electron heating for $f_0$ not near $sf_{\mathrm{e}}$ is consistent with our result of enhanced 630.0 nm emissions and electron temperatures for $f_0 = 6.200$ MHz.

Shindin et al. (2015) detected enhancements at 557.7 nm only for $f_0$ about 15–20 kHz above $4f_{\mathrm{e}}$ in experiments at the Sura facility, with the HF beam tilted 12° south from vertical in the magnetic meridional plane and the photometers directed in magnetic zenith (19° south). For a vertical HF beam, emissions were observed with $f_0$ about 5–15 kHz below $4f_{\mathrm{e}}$ and also at about 220–280 kHz above $4f_{\mathrm{e}}$. Our suggestion that the emission was enhanced for $f_0 \lesssim 4f_{\mathrm{e}}$ with the Heating beam in magnetic zenith thus agrees with Shindin et al. (2015) for a vertical Sura beam, but disagrees with their observations for the Sura beam tilted towards magnetic zenith. However, we do not find it clear whether Shindin et al. (2015) actually obtained conditions with $f_0 \lesssim 4f_{\mathrm{e}}$ and the HF beam tilted towards magnetic zenith, so that a comparison with the observed 557.7 nm enhancements for $f_0 \gtrsim 4f_{\mathrm{e}}$ could actually be made. Further, although Shindin et al. (2015) by superposed epoch analysis convincingly bring

forth the pump-enhanced 557.7 nm emission, their data remains noisy with variations in the natural background airglow. The experimental conditions in our case were better and the optical emissions were enhanced well above background. Another point is that even with $f_0$ about 15–20 kHz above $4f_e$ in some altitude region, enhancement of optical emissions could also occur at some ten kilometers or more lower altitudes at which $f_0 \lesssim 4f_e$.

Gustavsson et al. (2002) observed $I_{5577}/I_{6300} \approx 0.3$–0.4 for $f_0 = 4.040$ MHz which is near $3f_e$, in experiments with EIS-CAT Heating and an ERP of approximately 73 MW. From estimates of the reflection altitude, the authors found that $f_0 \lesssim 3f_e$ in the early part of the experiment which changed to $f_0 \gtrsim 3f_e$ in the later part because of the slowly increasing pump reflection height during the evening. We note that the largest ratio $I_{5577}/I_{6300} \approx 0.4$ was observed in the early part of the experiment (their figure 2), when $f_0 \lesssim 3f_e$, which is consistent with our results with $f_0 = 5.423$ MHz $\lesssim 4f_e$.

Kosch et al. (2005) presented experimental results from the HAARP facility and found $I_{5577}/I_{6300} \approx 0.2$–0.4, but for $f_0 \gtrsim 3f_e$ (their figure 1). We note that the largest ratio $I_{5577}/I_{6300} \approx 0.4$ was observed in the first displayed pump pulse when $f_0$ was the closest to $3f_e$. The ERP was 47.8 MW. However, no results were obtained for $f_0 \lesssim 3f_e$ to compare with. Thus, Kosch et al. (2005) observed the largest ratio $I_{5577}/I_{6300} \approx 0.4$ for $f_0 \gtrsim 3f_e$, while Gustavsson et al. (2002) observed their largest ratio $I_{5577}/I_{6300} \approx 0.4$ for $f_0 \lesssim 3f_e$. In addition, the ERP was larger in the experiment by Gustavsson et al. (2002) than for Kosch et al. (2005). One difference between these two experiments that may have influenced the excitation efficiency for the optical emissions is that Gustavsson et al. (2002) used a vertically transmitted pump beam while Kosch et al. (2005) transmitted the beam in magnetic zenith where maximum optical effects are expected (Kosch et al., 2000; Pedersen and Carlson, 2001; Pedersen et al., 2008). This could have contributed to the high ratio $I_{5577}/I_{6300}$ for $f_0 \gtrsim 3f_e$ observed by Kosch et al. (2005) despite the lower ERP.

## 6 Conclusions

The EISCAT Heating facility was used to excite ionospheric F region plasma by HF pumping 150 s on/85 s off and the beam pointing in geomagnetic zenith, first with $f_0 = 6.200$ MHz and thereafter with $f_0 = 5.423$ MHz. Optical emissions imaged at 557.7 and 630.0 nm from three ALIS sites were analyzed and plasma parameter values were obtained with the EISCAT UHF incoherent scatter radar. In addition, stimulated electromagnetic emissions were detected on the ground. From tomography-like reconstruction of the altitude distribution of the optical volume emission rates, we conclude that $f_0 = 6.200$ MHz $\approx 4.6f_e$ and $f_0 = 5.423$ MHz $\lesssim 4f_e$ in the height regions with the largest optical enhancements. The ratio of the column emission rates was $I_{5577}/I_{6300} \approx 0.2$ for $f_0 = 6.200$ MHz. As $f_0$ was decreased to 5.423 MHz, the pump-induced enhancements of both the 630.0 nm emission and electron temperature decreased. This is consistent with that the O($^1$D) state is excited mainly by electron heating, as reduced electron temperature means reduced heating. On the other hand, the emission intensity at 557.7 nm was approximately the same or even increased slightly as $f_0$ was decreased to 5.423 MHz, so that $I_{5577}/I_{6300} \approx 0.5$. Whereas emission at 630.0 nm has been attributed to excitation of the O($^1$D) state by mainly electron heating, emission at 557.7 nm from the O($^1$S) excited state is due to accelerated electrons. However, in view of the large ratio $I_{5577}/I_{6300} \approx 0.5$ and the importance of electron acceleration for the emissions at 557.7 nm it is reasonable that electron acceleration also contributed to the 630.0

nm emission. The obtained altitude profiles of the optical volume emission rates provide a limit on the wavelength range of upper hybrid waves considered to be instrumental for the electron acceleration. The 557.7 nm enhancement is suggested to depend primarily on upper hybrid waves with wavelengths of 1 m or longer that due to the linear dispersion characteristics can be excited for $f_0 = 5.423$ MHz $\lesssim 4f_e$. The fact that the growth times of both emissions in this case were several times longer than the lifetimes of their respective excited states, indicates that both emissions depend on the formation of small-scale density striations, which due to the limited upper hybrid wavelength range is expected to be slower for $f_0 \lesssim 4f_e$ than for $f_0$ not near $sf_e$ ($s \geq 3$).

*Data availability.* Access to the raw data may be provided upon reasonable request to the authors.

*Author contributions.* TBL organized the experiment, interpreted the results and wrote the paper. TS analysed the data, did the theoretical modeling and provided the plots for the figures. All took part in the experiment and contributed to the paper.

*Competing interests.* The authors declare that they have no conflict of interest.

*Acknowledgements.* The authors gratefully acknowledge the three referees, two anonymous and Savely M. Grach, for their valuable comments on the manuscript. EISCAT is an international association supported by research organizations in China (CRIRP), Finland (SA), Japan (NIPR and ISEE), Norway (NFR), Sweden (VR), and the United Kingdom (UKRI).

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
