# Peer review of "On mechanisms for HF pump-enhanced optical emissions at 557.7 and 630.0 nm from atomic oxygen in the high-latitude F-region ionosphere"

_Annales Geophysicae, 2023_

## Referee Comment (RC1)

Review report of "On mechanisms for HF pump-enhanced optical emissions at 557.7 and 630.0 nm from atomic oxygen in the high-latitude F-region ionosphere" by Leyser et al.

The main goal of this paper is to present the effects of HF pump-enhanced on optical emissions at 557.7 and 630.0 nm from atomic oxygen. The concept and analysis are sound and their results are important to the scientific community. However, the manuscript needs some improvements in order to clarify some point. Thus, I have some comments as follows:

**Major Comments:**

1. Just showing the results of only the effects of the pump-enhancement make it a bit difficult to follow. So I suggest the authors to include a control case, i.e., a case without pump-enhancement for all related results. In so doing the reader can see the visual differences between the ideal case and the enhanced case, thereby further clarifying the concept of the paper.

2. The introduction of the work is well written. However, it lacks the objective of the study, which obviously will give rise to the novelty of the work. I therefore suggest the authors clearly state the objectives of the work and the novelty of the research and how different this current work is from previous works. Therefore, the introduction and the conclusion should be updated.

3. I expect some information on the instruments to be given in the experimental setup section. It should not be assumed that the readers already know about the subject. Therefore, I encourage the authors to give a brief description of the instruments.

**Minor Comments:**

1. **Introduction**

**Line 15-16:** Why limiting the frequency $f_o$ to below or near the ionospheric critical frequency and with left-handed circular polarization? Besides the strongest effects obtained from these two consideration, are there other benefits?

**Line 102-104:** The emission intensity did not weaken as $f_o$ was decreased at about 16:50 UT. The images from the Kiruna site show even an increase in the emissions as $f_o$ was decreased, despite a lower ERP. But again the altitude region of the emissions decreased as $f_o$ was lowered.

Question: Are there any physical explanation to the characteristics of increasing emission but decreasing altitude with decreased $f_o$?

**2. Experiment setup**

Kindly include instrumentation.

**3. Experimental results**

Line 115: Please remove "." after "which" "……. , which. is ………

Line 115: Please add "of" between that and the in …. that the excitation ……

Note: I will suggest the entire sentence between Line 115 and 116 be rephrase.

**4. Emission growth times**

Line 144: Change "Further" to "Furthermore"

Line 168: Change "b oth" to "both".

**5. Discussion**

Line 269: With $f_0 \lesssim 4f_e$. to   With $f_0 \lesssim 4f_e$ (remove ".")

**6. Conclusion**

Lines 320-321: Kindly rephrase this sentence: "This is consistent with that the $O(^1D)$ state is excited mainly by electron heating".

I suggest the authors give a look at the conclusion and rephrase some sentences for clarity.

---

## Referee Comment (RC2)

The paper "On mechanisms for HF pump-enhanced optical emissions at 557.7 and 630.0 nm from atomic oxygen in the high-latitude F-region ionosphere" by T. B. Leyser, T. Sergienko, U. Brändström, B. Gustavsson, M. T. Rietveld presents interesting new results on artificial airglow at 630 nm (red line, level $O^1D$) and 557.7 nm ($O^1S$) excited by powerful HF radio waves at the EISCAT-heating. It should be noted that the results of different similar experiments are quite variable due to, first of all, variability of the ionosphere, and often it is difficult to find what parameters are responsible for different results. The presented paper contain very interesting new data, modeling and discussion which can be considered as a noticeable contribution to the artificial airglow study. So, the paper shall be published as a "regular paper".

Some aspects of the paper have to be clarified.

My comments/questions.

1.  Rows 174-176, the authors write:
    Excitation of the $O(^1D)$ state, the source of the 630.0 nm line, has been attributed to mainly electron heating from a maxwellian electron distribution (Mantas, 1994; Mantas and Carlson, 1996), however, taking into account collisional de-excitation by collisions with molecular oxygen and nitrogen in the atmosphere (Gustavsson et al., 2001).

Comment: In the paper V. V. Klimenko, S. M. Grach, E.N. Sergeev, A.V. Shindin, Features of the ionospheric artificial airglow caused by Ohmic heating and plasma turbulence-accelerated electrons induced by HF pumping of the SURA heating facility, Radiophysics and Quantum Electronics, Vol. 60, No. 6, November, 2017 (Russian Original Vol. 60, No. 6, June, 2017) DOI 10.1007/s11141-017-9812-0 it is shown that for the approximately same ERP the red line artificial airglow is attributed mainly to the electron acceleration, but not heating. This should be discussed in the paper. By the way, H. Carlson (passed away, unfortunately) agreed with conclusions of the latter paper (private conversation).

2.  Question: Authors compare the altitude of the volume emission rate maximum with the altitude where $f_0 \sim 4f_e$ (which is reasonable), but do not compare these altitudes with the pump wave upper hybrid altitude. Why? This altitude is known to be the altitude where the pump wave energy contribution to the ionospheric plasma is maximum.

3.  Rows 199-203. The authors write:
    Stimulated electromagnetic emissions can also be used to estimate the vicinity of $f0$ to $sf_e$ (Leyser, 2001), but in our experiment the emissions were generally too weak for spectral structure to be identified. However, for the fourth HF pulse at $f_0 = 5.423$ MHz, a weak broad upshifted maximum (BUM) can be identified. In Fig. 4 it can be seen that the electron temperature is slightly more enhanced in the fourth (and last) HF pulse than in the preceding pulses at $f_0 = 5.423$ MHz, which indicates a stronger plasma excitation in the last pulse. Etc.

Comment 1. The BUM peak position in the SEE spectrum strongly depends on $f0-4f_e$, so the appearance in the paper a figure with the SEE spectra would be reasonable.

Rows 315 -319:
From tomography-like reconstruction of the altitude distribution of the optical volume emission rates, we conclude that $f_0 = 6.200$ MHz $\sim 4.6f_e$ and $f_0 = 5.423$ MHz $< 4f_e$ in the height regions with the largest optical enhancements.

Comment 2. The absence of the SEE in the used pump wave frequencies can be related to belonging of these frequencies to specific ranges ("the weak emission range" for 6.2 MHz, and the "below harmonic range" (s=4) for 5.423 MHz, see Leyser, 2001 and Sergeev, E.N., Frolov, V.L., Grach, S.M., Kotov, P.V., 2006, On the morphology of stimulated electromagnetic emission spectra in a wide pump wave frequency range. Adv. Space Res. 38, 2518–2526). These ranges are known to have a weak SEE. Dependence of SEE spectra on $f_0-sf_e$ (which does not have an adequate interpretation yet) points to the

dependence of the efficiency of the PW energy contribution to the formation of the plasma wave spectra and efficiency of electron heating and electron acceleration on $f_0$-$sf_e$.

Finally. I am not agreeing with some points in the Section Discussion. However, this is not a barrier for the paper publication, but the question of further experiments and discussions.

---

## Author Response (AR1)

Dear topic editor,

Here is the uploaded revised version with tracked changes of the manuscript.

Best regards,
Thomas Leyser